# Morphological and Genetic Characterization of *Eggerthella lenta* Bacteriophage PMBT5

**DOI:** 10.3390/v14081598

**Published:** 2022-07-22

**Authors:** Sabrina Sprotte, Torben S. Rasmussen, Gyu-Sung Cho, Erik Brinks, René Lametsch, Horst Neve, Finn K. Vogensen, Dennis S. Nielsen, Charles M. A. P. Franz

**Affiliations:** 1Department of Microbiology and Biotechnology, Max Rubner-Institut, Federal Research Institute of Nutrition and Food, 24103 Kiel, Germany; gyusung.cho@mri.bund.de (G.-S.C.); erik.brinks@mri.bund.de (E.B.); horst.neve@mri.bund.de (H.N.); charles.franz@mri.bund.de (C.M.A.P.F.); 2Department of Food Science, Faculty of Science, University of Copenhagen, 1958 Frederiksberg, Denmark; torben@food.ku.dk (T.S.R.); rla@food.ku.dk (R.L.); fkv@food.ku.dk (F.K.V.); dn@food.ku.dk (D.S.N.)

**Keywords:** anaerobe, *Eggerthella lenta*, virulent phage, *Siphoviridae*, genome sequence

## Abstract

*Eggerthella lenta* is a common member of the human gut microbiome. We here describe the isolation and characterization of a putative virulent bacteriophage having *E. lenta* as host. The double-layer agar method for isolating phages was adapted to anaerobic conditions for isolating bacteriophage PMBT5 from sewage on a strictly anaerobic *E. lenta* strain of intestinal origin. For this, anaerobically grown *E. lenta* cells were concentrated by centrifugation and used for a 24 h phage enrichment step. Subsequently, this suspension was added to anaerobically prepared top (soft) agar in Hungate tubes and further used in the double-layer agar method. Based on morphological characteristics observed by transmission electron microscopy, phage PMBT5 could be assigned to the *Siphoviridae* phage family. It showed an isometric head with a flexible, noncontractile tail and a distinct single 45 nm tail fiber under the baseplate. Genome sequencing and assembly resulted in one contig of 30,930 bp and a mol% GC content of 51.3, consisting of 44 predicted protein-encoding genes. Phage-related proteins could be largely identified based on their amino acid sequence, and a comparison with metagenomes in the human virome database showed that the phage genome exhibits similarity to two distantly related phages.

## 1. Introduction

*Eggerthella* (*E*.) *lenta* is an obligately anaerobic bacterium which occurs in the gastrointestinal tracts of humans and animals [1,2]. It belongs to the class of the *Coriobacteriia* within the Gram-positive phylum *Actinobacteria*. Gupta et al. [3] proposed the division of species of the *Coriobacteriia* into two orders, i.e., the *Coriobacteriales* containing the families *Coriobacteriaceae*, *Atopobiaceae* and the order *Eggerthellales*, containing the *Eggerthellaceae* family and the genus *Eggerthella*. *Eggerthella* spp. are common members of the human gut microbiome, for instance, being detected in 81.6% of the individuals in a joint German and American cohort [4]. *Eggerthella* spp. were reported to occur in numbers of up to 7 × 10^5^ colony-forming units (CFU) g^−1^ in human feces, and appeared to be more associated to the gut wall than free living in the lumen [1]. *E. lenta* has been linked to positive effects in the host lipid metabolism [5]. Similar to other *Coriobacteriia*, *E. lenta* was also shown to be involved in the metabolism of bioactive secondary plant compounds such as resveratrol from grapes or daidzein from soy beans [6,7]. On the other hand, it has also been described as an opportunistic pathogen involved in bacteremia [8,9], to be more abundant in type-2-diabetes-patients [10,11] and to produce imidazole propionate that impairs insulin signaling [10]. The genome of strain *E. lenta* DSM 15644 used in this study has been previously sequenced and the bacterium was shown to be capable of the inactivation of the cardiac medication and plant natural product digoxin [4].

In the human gut, the community of phages (i.e., the phageome) has not been consistently assessed. For instance, Castro-Mejía et al. [12] and Liang et al. [13] found that human individuals harbor between 10^9^ and 10^10^ phage particles per gram of feces [12]. Another study estimated the number of virus-like particles between 10^10^ and 10^12^ per gram of feces [14]. In both cases, the number of prophages present in the genomes of gut bacteria was not yet included. The gut virome is highly diverse, individual-specific and stable over time [15]. It is suggested that 20–50% of the extracellular phages in the gut of healthy humans are represented by temperate phages [16,17], so they tend not to dominate [15]. Nevertheless, together with the prophages residing in the bacterial genomes, temperate phages constitute the majority of the phageome and this composition seems to be altered to more elevated levels of induced prophages in several diseases [18].

Despite their discovery dating over 100 years ago by d`Hérelle and Twort [19,20], the research on the identity of gut phages and on their effects on gut bacteria is still in its relatively early days [17,21,22]. Most studies on gut phages are based on metagenomic analyses of phages, rather than on the isolation of the phages themselves or knowledge of their hosts [15,23,24,25,26,27] One reason for this could be the lack of a clear standard method for isolating gut phages [28]. The few studies which so far have described the isolation of phages capable to form visible plaques under anaerobic conditions were focused on phages of the genera *Bacteroides* [25,29,30] and *Clostridium* [31], and the phages infecting the latter genera are almost exclusively temperate and were induced from the bacterial host genome. A member of the most abundant human-associated crAssphage group, phage ΦCrAss001, was only recently isolated for *Bacteroides intestinalis* [30]. A further reason for why the number of isolated gut phages is still quite low is that their bacterial hosts are mainly growing strictly anaerobically, and thus are difficult to cultivate and in general difficult to handle in the laboratory [17].

In our present study, we report on a phage screening performed with sewage, and strain *E. lenta* DSM 15644 was shown to be sensitive for a lytic bacteriophage. The study used the double-layer agar method which is commonly used for the isolation of phages and adapted it to anaerobic conditions, accounting for the fact that anaerobic bacteria usually grow to only low optical densities. Using this method, the *E. lenta* phage PMBT5 could be isolated. To our knowledge, PMBT5 is the first peer-reviewed published phage which can infect the obligate anaerobic intestinal bacterium *E. lenta*, although Soto Pérez reported the isolation of four *E. lenta* phages in a doctoral thesis (https://escholarship.org/uc/item/3kf0f1zn, accessed on 30 June 2022). Analysis of the genome revealed that PMBT5 represents a new type of phage within the *Caudovirales*, with only low sequence similarity to two other phages that were predicted from metagenomic data [25].

## 2. Materials and Methods

### 2.1. Media Preparation, Bacteria Strains and Cultivation

For preparation of the Wilkins–Chalgren (WC) Anaerobe Broth (CM0643; Oxoid, Munich, Germany) [32] used for anaerobic bacterial propagation, the recipe no. 339 of Deutsche Sammlung von Mikroorganismen und Zellkulturen (DSMZ, Braunschweig, Germany) was used. Briefly, 500 mL of WC Anaerobe Broth in a 750 mL flask supplemented with 500 µL L^−1^ resazurin sodium salt (0.1% *w*/*v*) as redox indicator was heated until boiling in a water bath to dissolve completely. The broth was kept in the boiling water bath until it became slightly pink, caused by an irreversible reduction of the resazurin to resorufin. Afterwards, the broth was flushed with 100% nitrogen (0.5 bar) for at least 30 min using either a hypodermic needle or a gas washing bottle while stirring (100 rpm) and cooling to room temperature. Finally, 0.3 g L^−1^ L-cysteine-monohydrochloride was added as reducing agent and the broth was aliquoted into 8 mL volumes using Hungate tubes (Ø 16 × 125 mm, Glasgerätebau Ochs, Bovenden/Lenglern, Germany), which were simultaneously flushed with nitrogen for a few seconds using a hypodermic needle. Hungate tubes were immediately closed with a bromobutyl rubber septum and a screw cap and sterilized by autoclaving. Alternatively, for lytic phage propagation on host bacteria at a large scale, 1 L WC Anaerobe Broth was prepared in a 1 L flask. After sterilization by autoclaving, the flask was transferred to the anaerobic workstation at least 24 h before use. In order to guarantee gas exchange, a cannula connected to a 0.2 µm sterile filter was inserted through the bromobutyl rubber stopper of the screw cap.

The WC top agar for the phage spot and plaque assays was prepared and stored in Hungate tubes as described above for the WC Anaerobe Broth, with the exception that 0.7% (*w*/*v*) agar was added before boiling, and finally aliquoting the top agar in 3 mL volumes in Hungate tubes. Those were stored refrigerated and gently boiled in a water bath before usage. The WC agar plates were prepared with 1.5% (*w*/*v*) agar, poured and dried under sterile aerobic conditions and then transferred, at least 24 h before use, into the anaerobic workstation. For longer anaerobic storage, e.g., if the bacteria needed several days to grow, the plates were placed in an empty petri dish bag together with moist paper towels to avoid drying out.

All bacterial cultivation steps were carried out in a Whitley A45 anaerobic workstation (Meintrup DWS, Herzlake, Germany) maintained with an anaerobic gas mixture (10% hydrogen, 10% carbon dioxide and 80% nitrogen). To passage the bacterial culture or to obtain fresh cultures for different tests, the bromobutyl rubber septa of the Hungate tubes were decontaminated with Chemgene^TM^ (Thermo Fischer Scientific, Waltham, MA, USA) or Virkon^®^ S (Meintrup DWS) before the cannula was inserted and about 300 µL culture were taken from the inverted Hungate tube and injected into a new prewarmed and similarly decontaminated Hungate tube. The *E. lenta* DSM 15644 strain used in this study was obtained from the DSMZ culture collection. *E. lenta* was grown for at least 16 h at 37 °C in WC Anaerobe Broth in Hungate tubes in the anaerobic workstation [32].

### 2.2. Bacteriophage Isolation, Propagation and Media

An anaerobic phage-isolation workflow, including anaerobic bacteria cultivation, as well as phage isolation and propagation, all adapted to anaerobic working conditions, is shown in Appendix A. Sewage from a municipal wastewater treatment plant near Kiel, Germany, was centrifuged (10 min, 10,000× *g*, 4 °C) and the supernatant was first filtered through a 4 to 7 µm pore size paper filter (S&S 595, Schleicher & Schuell, Germany) and finally through a 0.45 µm pore size membrane filter (Filtropur S, Sarstedt, Germany). Before use, a maximum of 10 mL filtrate was kept 24 h under anaerobic conditions in the anaerobic workstation (Appendix A). In order to obtain a concentrated *E. lenta* culture, 4 mL of overnight-grown culture (Appendix A) was harvested by centrifugating 2 times 2 mL in an Eppendorf reaction tube for 2 min at 10,000× *g*, both times discarding the supernatant. The cell sediments were resuspended in 100 µL of 100 mM CaCl_2_. For the enrichment of phages, 3 mL of the sewage filtrate were mixed with the 100 µL high cell-density, concentrated culture of *E. lenta* DSM 15644. After this, 10 mL prewarmed WC Anaerobe Broth were added and the sample was incubated for 24 h (Appendix A) in the anaerobic workstation. Following the phage enrichment step, the phages were then isolated using a double-layer agar method. For this, 100 µL of the filtered (0.45 µm) enrichment sample were spotted on an *E. lenta* DSM 15644 lawn consisting of concentrated bacterial cells as described above. These had been resuspended in 100 µL of 40 mM CaCl_2_, transferred to 3 mL molten WC top agar in a Hungate tube, vortexed and poured on a WC agar plate (Appendix A). Single plaques were isolated by three subsequent plaque assays with the concentrated bacterial culture applied as described above for the spot assay (Appendix A). Briefly, the top agar, which comprised the phage-derived lysis zone, was scraped off and resuspended in 1.5 mL SM buffer (0.58% NaCl, 0.25% MgSO_4_ x 7H_2_O, 0.24% Tris-HCl (pH 7.4) [33] without gelatine). This was then vortexed and left to stand for several hours in the anaerobic workstation. The SM buffer with the resuspended phages from the spot zone was aliquoted in reaction tubes and transferred to the anaerobic workstation 24 h before use. After filtration (0.45 µm), a serial ten-fold dilution of the phage lysate was prepared in SM buffer. A 100 µL volume of each dilution was combined with the concentrated bacterial culture and mixed by vortexing, followed by a 10 min incubation. Finally, one single plaque was diluted in 100 µL SM buffer [33] and used for lytic propagation in liquid culture (Appendix A). First, the optimal bacteria-to-phage ratio was determined at a small-volume scale in Hungate tubes. For this, 300 µL of an overnight grown *E. lenta* culture were mixed with 100 µL 40 mM CaCl_2_ and 100 µL of serial ten-fold dilutions of phage lysate (10^−1^ to 10^−3^) in a Hungate tube filled with 8 mL WC Anaerobe Broth. The optical density (OD_620nm_) was determined for the first time after 4 h and again every 2 h later. In the subsequent experiments, the amount of (i) the optimal phage/lysate dilution, by which complete bacteria lysis was achieved, and of (ii) CaCl_2_ were extrapolated to a 10 mL overnight grown *E. lenta* culture, which was incubated for 10 min. Then, the complete sample was transferred to 1 L of anaerobic, prewarmed WC Anaerobe Broth. The culture was then incubated for nearly 48 h while stirring at 100 rpm (Appendix A) at 37 °C in the anaerobic workstation. The phages were purified from the supernatant by cesium chloride (CsCl) density gradient ultracentrifugation (Appendix A) as described previously [33].

### 2.3. Transmission Electron Microscopy

Phages were adsorbed to a freshly prepared ultra-thin carbon film and were fixed with 1% (*v*/*v*) EM-grade glutaraldehyde for 20 min. After negative staining with 1% (*w*/*v*) uranyl acetate, specimens were analyzed using a Tecnai 10 transmission electron microscope (FEI Thermo Fisher Scientific, Eindhoven, The Netherlands) at an acceleration voltage of 80 kV. Digital micrographs were captured with a MegaView G2 CCD-camera (EMSIS, Muenster, Germany).

### 2.4. Phage DNA Preparation

For phage DNA isolation, the Quick-DNA^TM^ Fungal/Bacterial MiniPrep Kit (Zymo Research, Freiburg, Germany) was used with some modifications. Briefly, 1 mL of high titer (~1 × 10^11^ plaque-forming units (PFU) mL^−1^) phage lysate obtained from CsCl density gradient ultracentrifugation was treated with DNase and RNase (each 20 µg mL^−1^) for 3 h at 37 °C. Inactivation of the enzymes was performed for 5 min at 75 °C. Then, 1 mL lysis solution from the DNA isolation kit was added and mixed thoroughly for 10 s. Proteinase K was then added at a final concentration of 80 µg mL^−1^ and the sample was incubated first for 30 min at 55 °C, then at 65 °C for 15 min while inverting the sample 2–3 times. The sample was mixed with 640 µL isopropanol and loaded onto a DNA-binding column from the kit. From this point on, the protocol of the manufacturer was followed. Finally, DNA amounts were quantified using a Qubit 3 fluorometer (Invitrogen, Darmstadt, Germany).

### 2.5. Genome Sequencing and Phylogenetic Analyses

Phage genome sequencing was performed using the TruSeq Nano DNA LT library Preparation Kit (Illumina, Munich, Germany) and the MiSeq Reagent Nano Kit v2 (Illumina) (500 cycles) according to the manufacturer’s instructions. Paired-end sequencing was performed on an Illumina MiSeq sequencer with 2 × 251 cycles. The MiSeq^®^ Reporter software was used for base calling, demultiplexing and adapter trimming directly on the MiSeq sequencer. The reads were assembled de novo into contigs with SPAdes 3.11.1 [34], followed by annotation of open reading frames (ORFs) on the contigs with PATRIC 3.5.29 [35]. This service uses the RAST tool kit (RASTtk) [36]. Locations of predicted ORFs were compared to those predicted by GeneMarkS [37] and annotated amino acid sequences were manually compared to those suggested by BlastP [38] and HHPred [39].

All pairwise comparisons of the nucleotide sequences were conducted using the genome-BLAST distance phylogeny (GBDP) method [40] with settings recommended for prokaryotic viruses [41]. The resulting intergenomic distances were used to infer a balanced minimum evolution tree with branch support via FASTME including superconvergent patch recovery (SPR) postprocessing. The branch lengths are scaled in terms of the GBDP distance formula d0 [42]. Branch support was inferred from 100 pseudo-bootstrap replicates each. Trees were rooted at the midpoint [43] and visualized with FigTree v. 1.4.3 [44]. Taxon boundaries at the species, genus and family level were estimated with the OPTSIL program [45], the recommended clustering thresholds [33] and an F value (fraction of links required for cluster fusion) of 0.5 [46]. ELM phage genomes [25] (*E. lenta* phages computationally predicted from metagenomes) were derived from (https://github.com/jbisanz/HuVirDB, accessed on 15 October 2021) and PMBT5 (GenBank acc. no. MH626557.1). Genome alignment and calculation of identity percentage was performed with EMBOSS Stretcher [47].

### 2.6. Phage Structural Protein Analysis by UHPLC-MS-MS

A filtered (0.45 µm PES syringe filter) phage stock (~5 × 10^9^ PFU mL^−1^) from liquid media was concentrated with Centriprep containing a 30 kDa filter and centrifuged at 1500× *g* until the volume was reduced to 4 mL, then diluted with SM buffer without gelatine and centrifugation was repeated three times, and the final volume was 2 mL phage suspension. The dialyzed and concentrated phage particles (~1 × 10^10^ PFU mL^−1^) were used for the assays. Phage proteins were identified as described previously [48]. Briefly, 75 µL of dialyzed and concentrated phage solution were mixed with 75 µL 1% SDS and incubated (30 min, 80 °C) before TCA precipitation. The proteins were re-solubilized in 8 M urea, 45 mM DTT and 50 mM Tris, pH 8.0, reduced and alkylated and then digested by trypsin (0.1 µg µL^−1^, Sigma Aldrich, Søborg, Denmark). The resulting peptides were analyzed using a Vanquish Flex Binary UHPLC system (Thermo Fisher Scientific, Hvidovre, Denmark) with an Aeris PEPTIDE 1.7 µm XB-C18, 150 × 2.1 mm column (Phenomenex, Værløse, Denmark) coupled with an Orbitrap Exploris 480 mass spectrometer (Thermo Fisher Scientific, Hvidovre, Denmark). The data were analyzed with Proteome Discover (version 2.3, Thermo Fisher Scientific) using a homemade protein database based on the obtained DNA sequence ORF predictions. Results were filtered in Proteome Discover with the integrated Target decoy PSM validator algorithm to a *q*-value of <0.01, which ensures a peptide–spectrum match false discovery rate below 0.01.

## 3. Results and Discussion

### 3.1. Isolation Protocol of E. lenta Phage PMBT5 under Anaerobic Conditions

There are only few studies describing successful methods of virulent phage isolation for intestinal bacteria under anaerobic conditions [29,30]. One reason for this could be the lack of a suitable isolation standard [28] and of appropriate isolation protocols. One of the aims of this study, therefore, was to provide a detailed description of phage isolation and to optimize the methodology for isolating lytic phages for anaerobic bacteria.

Phage PMBT5 was isolated from sewage obtained from a municipal wastewater treatment plant. The processing of such highly diluted liquid samples is not time consuming, as the suspended solids can be removed by a few centrifugation steps and the supernatants can easily be filtered through a 0.45 µm pore size membrane filter to separate the phages from most of the bacteria. Furthermore, the phage titer in sewage has been reported to be moderately high, i.e., from 2 × 10^7^ PFU mL^−1^ [49] up to 10^9^ virus particles mL^−1^ [50], so that a concentration step by, e.g., PEG precipitation or CsCl ultracentrifugation is not necessary. For the far more demanding phage isolation directly from human feces, the protocols according to Shkoporov et al. [30], Castro-Mejia et al. [12], Deng et al. [51] or Mathieu et al. [52] may be used.

Another reason why so far only few phages infecting strictly anaerobic intestinal bacteria have been isolated and characterized may be their challenging cultivation conditions, as many of these bacteria do not grow at all, or grow only poorly, under anaerobic laboratory conditions using general microbiological culturing media. However, a dense bacterial lawn on an agar plate is a prerequisite to detect phage-derived plaques that can be picked for phage isolation and purification. An easy to handle way to obtain a sufficient lawn with bacteria that grow only slightly turbid under anaerobic conditions was used in this study. To achieve this, 4 mL of anaerobic bacterial culture was concentrated by centrifugation, yielding a bacterial suspension with an optical density (OD_620nm_) of nearly 1. This concentration of bacteria may be critical for the identification of lysis zones or plaques caused by phages, and possibly even for the enrichment of the phage from the environmental sample. The intestinal bacterium used in this study, *E. lenta* DSM 15644, routinely reached an OD_620nm_ of approximately 0.3 in the WC Anaerobe Broth after 24 h incubation. Other gut bacteria, e.g., *Faecalibacterium prausnitzii,* also showed limited growth in liquid cultures in a previous study [53]. Furthermore, bacteria growing sufficiently in liquid broth may not grow well on agar plates and *vice versa*. In these cases, our protocol may be adapted accordingly, as several milliliters of bacteria culture can be used for the concentration or, alternatively, several bacterial colonies can be picked from the agar plates and resuspended in 100 µL CaCl_2_ (either 40 mM or 100 mM). In addition, as required for all materials and reagents used for the isolation of phages from anaerobic bacteria, the top agar (which can only be liquified by boiling and subsequent incubation at temperatures of at least 50 °C in a water bath) should also be anaerobic. Therefore, Hungate tubes were chosen to prepare and store the top agar anaerobically until use. The advantages were that they could (i) be sealed airtight, (ii) be boiled in a water bath and were (iii) relatively easy to transfer into the anaerobic workstation (especially via a cuff-less entrance). Phage PMBT5 was isolated from plaques which appeared in top agar that was inoculated with *E. lenta* DSM 15644 as described above at 37 °C incubation under anaerobic conditions. Phage PMBT5 was also tested for lytic activity against a limited number of strains available in the laboratory, i.e., *Eggerthella lenta* DSM 2243, *Slackia isoflavoniconvertens* DSM 22006 and *Slackia equolifaciens* DSM 24851, but no plaques were obtained with these strains (results not shown).

### 3.2. Morphology of Phage PMBT5 Particles and Phage Plaques

The morphology of phage PMBT5 was analyzed from the CsCl density gradient purified lysate by transmission electron microscopy, and 14 phage particles were measured in detail. The phage showed an isometric head (diameter 56.1 ± 1.9 nm) and a flexible, noncontractile tail (length incl. baseplate: 128.1 ± 2.6 nm, width: 10.3 ± 0.3 nm), indicating that phage PMBT5 could be assigned to *Siphoviridae*-like phages (Figure 1). It possessed a distinct baseplate (width: 18.1 ± 2.0 nm) with a protruding tail fiber (length: 44.8 ± 2.2 nm) with typical cylindric extensions at the distal ends. 

A critical step for the purification of phages is to obtain enough phage lysate with at least a moderate titer. In our case, 1 L of phage PMBT5 cocultured with *E. lenta* DSM 15644 reached about 10^9^ PFU mL^−1^ after 48 h incubation and was sufficient to obtain 1 × 10^11^ PFU mL^−1^ CsCl purified phages.

Plaque formation was analyzed after 24 h incubation using the double-layer agar method with WC Anaerobe Broth adapted to anaerobic conditions (Figure 2). The plaques were slightly turbid and ranged from 1 to 2 mm in diameter and were surrounded by a halo (diameter 4 mm), i.e., semitransparent zones around the plaques. These haloes were hypothesized to possibly result from the diffusion of soluble, phage-produced enzymes (e.g., EPS depolymerases) destroying the cell envelope and leading to diffuse, semitransparent zones [54,55,56]. As the genes encoding these enzymes could not be identified in the phage PMBT5 genome, the turbid haloes surrounding the plaques may also be explained by the excess lysin production by the phage PMBT5. As visible in Figure 2, *E. lenta* DSM 15644 was able to grow densely as bacterial lawns on the agar.

### 3.3. Phage PMBT5 Genome Sequence and Phylogenetic Analysis

The genome of PMBT5 consisted of a linear double-stranded molecule (dsDNA) of 30,930 bp and a mol% GC content of 51.3, which is lower than the mol% of 64.2 of the genome of *E. lenta* DSM 15644. A total of 34,093 reads were assembled into one contig using SPAdes 3.11.1 with an average coverage of 529 times. The annotation process using PATRIC [35] led to 44 putative open reading frames (ORFs) (Appendix A, Figure 3), which were all orientated in the same direction.

The HHpred software was mainly used to predict the protein functions of phage PMBT5 in terms of the DNA packaging module, capsid, head-to-tail joining and tail module, adhesion device, lysis cassette and replication module. For the DNA packaging module, ORF2 from phage PMBT5 matched significantly large subunit terminases from various phages. Based on BlastP and SMART [57] analyses, the terminase large subunit revealed three conserved domains, i.e., phage_term_2 (PBSX family), terminase_6 and XtmB. Those domains have also been found in the following *pac*-type phages: *Shigella* phage Sf6, *Lactobacillus* prophage Lj965, *Escherichia coli* phage T4 and *Salmonella* phage P22, suggesting that phage PMBT5 uses the ‘headful’-mechanism for DNA packaging as well. However, using PhageTerm to analyze the mode of packaging, the analysis indicated an unknown mode of packaging. No matches were obtained for ORF1; hence, a small subunit function cannot be predicted. However, we noted that ORF1 shows similar gene syntheny (position on the genome) and deduced protein size (in terms of number of amino acids) as other terminases of the *Siphoviridae*-type phages.

A number of PMBT5 ORFs matched with various phage-encoded structural proteins of the capsid and head-to-tail joining module: ORF4 has homology to portal proteins from 14 phages (highest probability match with *Bacillus subtilis* phage SPP1) [58]. The ORF4 structural protein was also identified by mass spectrometry (Table 1). ORF16 was also identified as the major capsid protein MCP of PMBT5 by mass spectrometry and HHPred analysis. Matches with bacterial head-to-face interface/DNA gatekeeper proteins were obtained for ORF17 and ORF18, with high similarity to corresponding genes of a *Bacillus* defective prophage element PBSX [59,60] and *Bacillus* phage SPP1 [61]. ORF20 has similarity to a tail-to-head joining protein of phage SPP1. ORF19 and ORF20 display a one base overlap, which is also known from other phages as a frame shift motive leading to translational coupling, e.g., gpG in lambda (PMBT ORF19) and gpGT in lambda (ORF19_ORF20 fusion protein of PMBT5) and lactococcal phages as well [62,63]. A coiled coil domain was suggested for ORF5 that matched a methyltransferase subunit of a type I restriction–modification system of different bacteria in the N-terminal region [64], which are also known to occur in bacteriophage genomes [65]. Finally, ORF9 showed homology to repeat five residue (Rfr) proteins of various bacteria [66]. By HHPred analysis, the major tail protein of phage PMBT5 could be linked to ORF21, which matched with four phages; coliphage lambda [67], lactococcal phage TP901–1 [68], *Staphylococcus aureus* phage 80alpha [69] and *Bacillus* phage SPP1 [70,71]. A similar function of ORF21 was also suggested by mass spectrometry (Table 1).

For *Siphoviridae* phages, the distal end of the tail is crucial for the recognition of the cell surface receptors for subsequent binding and DNA ejection. The most important structural proteins are (part of) the tape measure protein (TMP), the distal tail protein (DIT) and the tail associated lysin (TAL), as well as the receptor binding protein (RBP) and, in many cases, further ancillary structural proteins. Typically, the *tmp* gene is followed by the *dit* and *tal* genes, and this complex is described as the TMP-DIT-TAL “triad” [72] as the key components of the phage adhesion device. A corresponding complex of phage PMBT5 could be characterized by HHPred analysis and was further supported by mass spectrometry. The HHPred analysis showed homology of ORF24 (with transmembrane domain) exclusively to the tape measure protein of the *Staphylococcus aureus* phage 80alpha [73]. ORF25 showed high homology to the DIT (N-terminal region only) of phage 80alpha and furthermore to the corresponding DIT proteins of *Bacillus cereus* phage SPP1 [74] and to two lactococcal phages p2 [75] and TP901–1 [76]. ORF26 of phage PMBT5 matches with a distinct fiber lower protein (FibL, gp62) of *S. aureus* phage 80alpha, and six of these short FibL fibers are coating/decorating the baseplate complex of phage 80alpha and are probably involved in host interaction [73]. The PMBT5 ORF26 protein could also be identified by mass spectrometry (Table 1). Corresponding short tail fibers are also visible on the micrographs of PMBT5, as indicated in Figure 1 (see triangles in right picture). Conformational changes have been reported for the phage 80alpha baseplate (including the FibL fibers) [73], and this flexibility of the baseplate components may also explain why the PMBT5 fiber structures are not visible on all micrographs. ORF27 of PMBT5 (with coiled coil domain) matches in the N-terminal region with the tail associated lysin (TAL) protein of phage 80alpha [73], indicating that this structural protein is part of the baseplate and/or of the single long tail fiber of phage PMBT5.

While ORF24, ORF25, ORF26 and ORF27 clearly match with the structural baseplate proteins of *S. aureus* phage 80alpha, unique features were predicted for ORF28. The HHPred analysis revealed both coiled coil and transmembrane domains, which matches with high probability with the tail needle proteins of podoviruses of Gram-negative bacteria, e.g., *Salmonella* phages HK620 and P22 (gp26) [77,78], and others. The P22 gp26 in the alpha-helical coiled coil conformation forms the (exposed approx. 12 nm) long tail fiber and is required for host cell attachment and subsequent penetration through the cell envelope. Tail needle proteins are well conserved among the taxonomically distinct P22-like phages (*Podoviridae*); therefore, it is notable that a putative tail needle structural protein could be identified in phage PMBT5 as well. The putative tail needle protein of PMBT5, however, is noticeably longer (45 nm) than those of the P22-like phages [77,78]. A striking similarity has been shown for the crystal structures of the tail needle protein of podoviruses and the (much shorter) receptor binding protein of lactococcal phage TP901–1 [79]. The C-terminal part of the gp26 gp of podovirus P22 is helical, but those of the tail needle protein of podovirus Sf6 and the receptor binding protein (RBP) of phage TP901–1 form typical knobs. The distal part of the phage PMBT5 tail spike has a cylindrical extension (Figure 1), and we speculate that this part is also functioning for the host receptor binding. The putative PMBT5 tail needle protein could also be identified by mass spectrometry (Table 1). In summary, the putative adhesion device of phage PMBT5 is strikingly complex and different from other *Siphoviridae* phages. It also reveals a modular structure with ORFs 24–27 (TMP, DIT FibL, TAL) resembling the corresponding structures of *S. aureus* phage 80alpha but, on the other hand, with ORF28, with a similarity to the tail needle proteins of podoviruses of Gram-negative bacteria.

Phage endolysins are peptidoglycan-specific hydrolases required for host–cell lysis after phage infection and reveal various complex forms, including catalytic domains, cell wall binding domains and nonbacteriolytic endopeptidases [80]. HHPred was also used to elucidate the lysis cassette of phage PMBT5. ORF30 of PMBT5 clearly indicated a multiple domain architecture of a putative endolysin. In the N-terminal part, ORF30 matches with N-acetylmuramoyl-L-alanine amidases of Gram-positive bacteria (*Streptococcus pneumonia*, *Clostridium intestinale* and others) and of phages, e.g., *Staphylococcus* phage GH15 [81]. The predicted C-terminal domain, however, is notable, as it matches exclusively with dextran (alternan) sucrases of lactic acid bacteria (*Limosilactobacillus fermentum*, *Leuconostoc mesenteroides* and other LAB) [82] (Appendix A), while there were no indications for the cell wall binding domains or the bacteriolytic endopeptidase domains. To our knowledge, such an uncommon multidomain architecture has not been reported before for endolysins of phages of Gram-positive bacteria and the biological function is not clear so far but may hypothetically relate to cell wall binding. A holin gene is typically present upstream of a phage endolysin gene. A good putative candidate might be ORF29 on the PMBT5 genome, but the HHPred analysis does not support this assumption, as no matches were obtained. The only evidence is the presence of a transmembrane domain, as holins are a group of (small) pore-forming membrane proteins required for the controlled access of phage endolysins to the peptidoglycan [83].

The HHPred analysis also allowed the identification of several PMBT5 ORFs from the replication module [84], i.e., ORF34 (DNA gyrase inhibitor), ORF35 (single-stranded binding protein), ORF39 (replication initiator protein, DnaD-like), ORF40 (primosomal protein, DnaI-like) and ORF41 (repressor/transcription regulator). ORF43 of phage PMBT5 is remarkable, as HHPred predicted a RuvC (Holliday junction resolvase), which are widely spread in Gram-negative bacteria [85] and are only in few cases known for phages of Gram-positive bacteria, i.e., lactococcal phage bIL67 [86]. For ORF44 (with a predicted coiled coil domain), significant matches were obtained for bacterial MazG-like nucleotidpyrophosphohydrolases. These enzymes can hydrolyze all eight canonical ribo- and desoxynucleoside-triphosphates to their respective monophosphates and PP(i) [87,88].

HHPred also predicted an ARNA transcription and a zinc binding protein (ORF7). Furthermore, a minor capsid protein (ORF8 and ORF19), a Lar family restriction alleviation protein (ORF11), a SHOCT domain-containing protein (ORF12) and a minor structural protein (ORF15) were identified by BLASTP only (but not by HHPred) at varying levels of identity but generally with high probability scores (i.e., low associated *e*-values).

Six ORFs were identified as structural proteins by mass spectrometry. The portal protein (ORF4), also involved in genome packaging, was identified as a structural protein based on one peptide, while the major capsid protein (ORF16) was identified by 10 peptides. The major tail protein (ORF21) was identified by three peptides, while the three hypothetical proteins probably involved in forming the distal tail protein (ORF25), the fiber lower protein (ORF26) and the tail needle protein (ORF28) were identified by one peptide each (Table 1). The functions of these hypothetical proteins were inferred by HHPred. The low number of identified structural proteins is possibly due to the low titer of the dialyzed and concentrated stock used.

Using ResFinder 3.1 [89] to search for antibiotic resistance genes, no such genes could be found. Furthermore, the absence of virulence factors was confirmed with PATRIC [35]. Those are important prerequisites for using the phage for biocontrol or phage therapy applications. Although, the phage was nonconfidentially (i.e., averaged probability of 0.531) predicted as having a temperate lifestyle [90], the manual search showed no evidence for lysogenic cycle-promoting genes such as phage integrase, repressor and antirepressor genes. Thus, the data strongly supported a lytic lifestyle of phage PMBT5.

PMBT5 possessed no tRNA-encoding genes in its genome when analyzed using tRNAscan-SE 2.0 [91]. Analyzing the host genome for the presence of prophages, we found several potential phage genomes that were predicted to be active and with a score above 0.90 using Prophage Hunter [92]. Recently, 13 putative *E. lenta* phages were predicted on basis of metagenome data [25]. To evaluate the validity of these so-called ELM phages, we performed a phylogenomic analysis based on the genome-wide BLAST distances of PMBT5, which predicted PMBT5 to belong to the same phage family as ELM P3 and ELM P10 (Figure 4). The genomic identities of PMBT5 compared to ELM P3 and P10 were 47.1% and 48.5%, respectively.

## 4. Conclusions

In this study, a method to isolate lytic phages from an anaerobic intestinal *E. lenta* strain was developed which may facilitate further studies on the intestinal phageome. We isolated a putative virulent phage (PMBT5) infecting the gut microbe *E. lenta.* Phage PMBT5 is a potential candidate for the modulation of the gut microbiota with regard to *E. lenta*, as the phage shows a lytic infection cycle and harbors neither virulence nor antibiotic resistance genes. To investigate this potential further, extensive host range studies would be required.

## Figures and Tables

**Figure 1 viruses-14-01598-f001:**
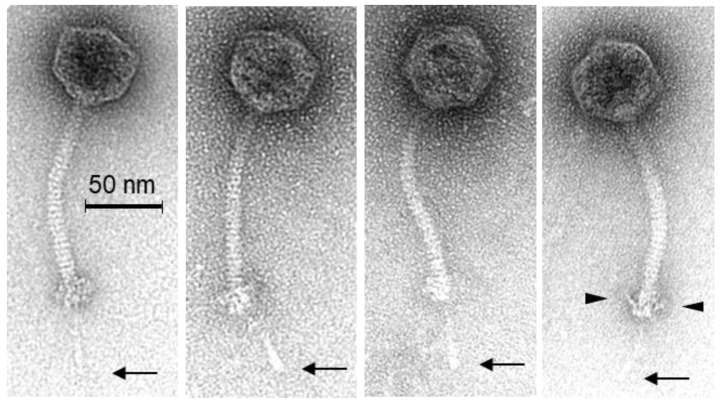
Transmission electron micrographs of *Eggerthella lenta* phage PMBT5 stained with 1% (*w*/*v*) uranyl acetate. The thin arrows indicate the central tail fiber under the baseplates with typical cylindrical extension at the distal end. The triangles show protruding baseplate appendages in upwards folded position.

**Figure 2 viruses-14-01598-f002:**
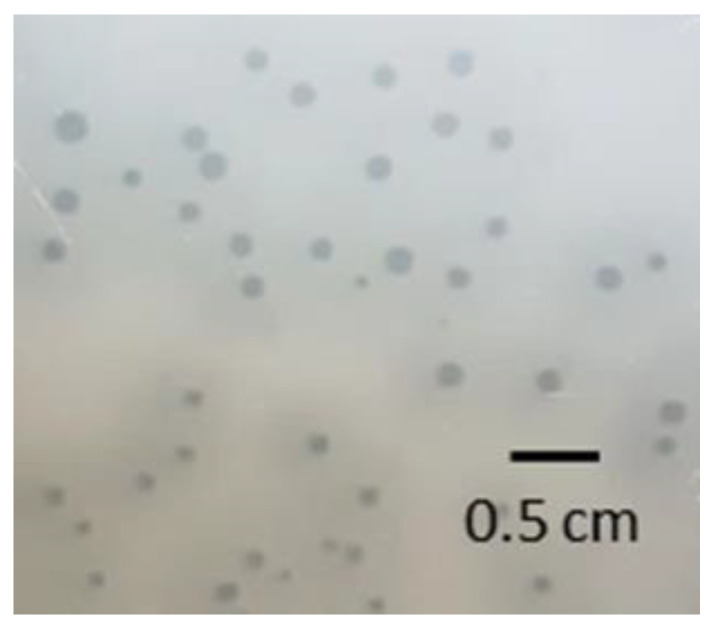
Phage PMBT5 plaques formed on an *Eggerthella lenta* DSM 15644 lawn in Wilkins–Chalgren Anaerobe Broth after 24 h incubation under anaerobic conditions. The plaques were lightly turbid and ranged from 1 to 2 mm in diameter and were surrounded by a halo (max. diameter 4 mm).

**Figure 3 viruses-14-01598-f003:**
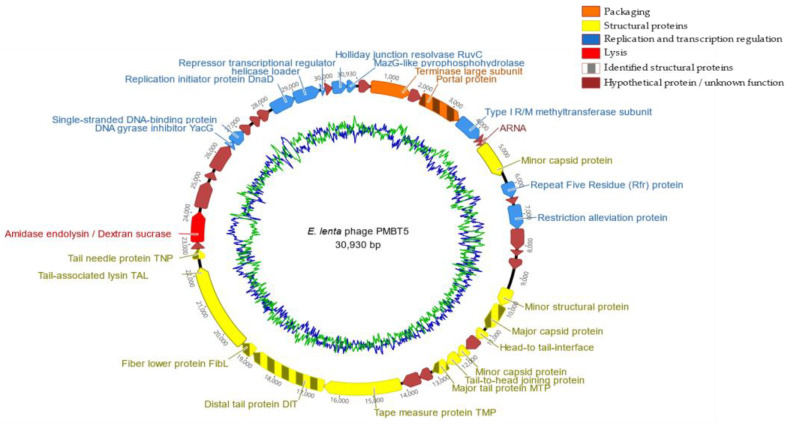
Genome map of *E. lenta* infecting phage PMBT5 with mol% GC content shown as an inner circle. ORFs are color-coded according to their predicted function. The genome was subdivided into functional modules as demonstrated by colors (for details see legend). The genome starts with the ORF upstream of the putative terminase large subunit (ORF2) shown at the top of the map. The map was generated using Geneious version 11.0. The genome is displayed here as circular molecule for a better overview, although the molecule is actually linear. The green line represents the mol% AT content while the blue line represents the mol% GC content. Striped ORFs indicate that the corresponding deduced proteins were confirmed by mass spectrometry.

**Figure 4 viruses-14-01598-f004:**
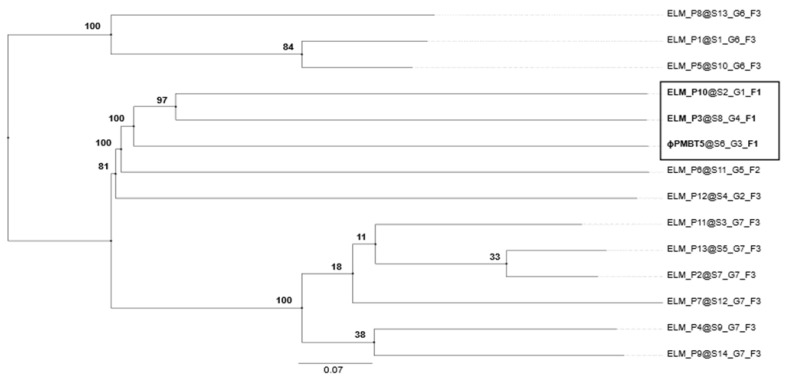
Phylogenomic tree of ELM phages [25] (metagenomicly predicted *E. lenta* phages) and PMBT5 based on genome-wide BLAST distances with an average support of 69%. Numbers on the tree represent pseudo-bootstrap values from 100 replications. The black square marks the ELM phages that are predicted to be within the same family (F1) as PMBT5. S = species, G = genus, F = family, and associated numbers indicate predicted groupings.

**Table 1 viruses-14-01598-t001:** Identified ORFs as structural proteins by UHPLC-MS-MS.

Accession *	Description	Coverage [%]	No. of Distinct Peptides Identified	No. of Amino Acids	Molecular Weight (kDa)
lcl_gp04 **ORF4** **Portal protein**	[locus_tag = HOT76_gp04] [db_xref = GeneID:54998161] [protein = portal protein] [protein_id = YP_009807283.1] [location = 2003..3385] [gbkey = CDS]	2	1	460	51.3
lcl_gp16 **ORF16** **Major capsid protein**	[locus_tag = HOT76_gp16] [db_xref = GeneID:54998173] [protein = main capsid protein] [protein_id = YP_009807295.1] [location = 10024..10899] [gbkey = CDS]	31	10	291	31.6
lcl_gp21 **ORF21** **Major tail protein**	[locus_tag = HOT76_gp21] [db_xref = GeneID:54998178] [protein = hypothetical protein] [protein_id = YP_009807300.1] [location = 12569..13075] [gbkey = CDS]	23	3	168	18.1
lcl_gp25 **ORF25** **Distal tail protein**	[locus_tag = HOT76_gp25] [db_xref = GeneID:54998182] [protein = hypothetical protein] [protein_id = YP_009807304.1] [location = 16515..18968] [gbkey = CDS]	1	1	817	89.8
lcl_gp26 **ORF26** **Fiber lower protein**	[locus_tag = HOT76_gp26] [db_xref = GeneID:54998183] [protein = hypothetical protein] [protein_id = YP_009807305.1] [location = 18968..19426] [gbkey = CDS]	7	1	152	16.3
lcl_gp28 **ORF28** **Tail needle protein**	[locus_tag = HOT76_gp28] [db_xref = GeneID:54998185] [protein = hypothetical protein] [protein_id = YP_009807307.1] [location = 22521..22841] [gbkey = CDS]	9	1	106	11.8

* In bold: Deduced functions from HHPred analysis.

## Data Availability

The genome of phage PMBT5 has been deposited in the National Center for Biotechnology Information (NCBI) under the GenBank accession no. MH626557. The version described in this paper is the version MH626557.1.

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
