# Peer review of "Morphological and Genetic Characterization of Eggerthella lenta Bacteriophage PMBT5"

_viruses, 2022, doi:10.3390/v14081598_

Round 1

Reviewer 1 Report

Sprotte et al describe the isolation of a bacteriophage that infects E. lenta, a bacteria commonly found in the gut microbiota. This paper is of interest because of the lack of characterized phages from the gut microbiota and specifically no phages have been isolated that infect E. lenta up until this point. The authors develop a technique to allow the isolation of phage from anaerobic, difficult to culture bacteria. This, in turn, allows them to isolate and characterize a phage for E. lenta. The phage is sequenced and the phage particles are analyzed using UHPLC-MS-MS. Also the phage is bioinformatically compared to other related phages. Overall the characterization of the phage and techniques used to isolate it is compelling and should be of interest to researchers in the field.

Minor points:

At line 273 the transition from the technique used to grow the bacteriophage in E. lenta and the description of PMBT5 is abrupt. I think a small description of how PMBT5 was isolated would be beneficial apart from what is in the methods. 

PMBT5 is called virulent throughout the text. This language should be softened as there are no experiments demonstrating that. While I agree, it probably does not integrate based on the phage proteins identified, it could be maintained episomally. 

In Figure 3 it is unclear what the green and blue lines represent. 

Reviewer 2 Report

The manuscript reports a classical phage description og phage PMBT5 from Eggerthella lenta. The report contains different sections concerning phage sequencing, propagation and annotation, all very standard feature. On top of this, the phage structure analysis reports remarkable nsEM images of the phage. they indicate that this phage belongs to Siphoviridae, with a rather short tail as the TMP is also rather short. However, the authors failed to interpret the remarkable features of these image. This is due to a poor, almost inexisting annotation of the most interesting part of phages, their adhesion device! This adhesion device is know to be located between the TMP and the holin/lysin. Using blast to annotate it is hopless, but recently, use of HHpred and even more recently of AlphaFold2 allows to annotate safely the adhesion device. If the authors had done it, they could easily assign the lateral extensions and what they call the « tail fiber » to well known component and explain why these proteins are so much visible; and maybe suggest a attachment mode of this phage to its host.

Reviewer 3 Report

Sprotte et al. discuss the isolation and characterisation of a novel phage infecting the gut bacterium Eggerthella lenta

I have several issues with the manuscript, which are discussed below

Major comments

  • Line 81: Turnbaugh Lab has a PhD thesis available online describing E. lenta phages (https://escholarship.org/uc/item/3kf0f1zn)
  • Line 244-246: The number of phages in sewage water is discussed, and it not being necessary to concentrate samples by PEG or CsCl ultracentrifugation. But where E. lenta phages not enriched (Lines: 137-143) by incubation with host bacterium prior to plating onto overlay for their detection in said material?
  • Line 308-309: Where sequence reads aligned back to the genome of phage to determine if regions of increased reads depth could be detected? Detection of such regions indicates the presence of terminal repeats on the genome.

Description of such finding in below paper:

Things Are Getting Hairy: Enterobacteria Bacteriophage vB_PcaM_CBB (10.3389/fmicb.2017.00044)

  • Line 355 – 356: Recent article described the prophages of E. lenta, which is likely worth mentioning

Selective Isolation of Eggerthella lenta from Human Faeces and Characterisation of the Species Prophage Diversity (doi.org/10.3390/microorganisms10010195)

  • Line 362 – 363: Is there any gene synteny between the genome of PMBT5 and metagenomic derived genomes (ELM P3 and ELM P10)? Do these metagenomic derived genomes of similar sizes? Or do they look incomplete? Is there any interesting difference in gene content between genomes or any interesting gene conservation?
  • Phage PMBT5 is indicated as a potential candidate for the modulation of the gut microbiota in the article. Did phage show board host range on a panel of different strains of this bacterium? Did phage show any potential to cause observable lysis of E. lenta in broth or inhibition of its growth? Such information should be stated if mentioned claim is made in the article.
  • Line 322 – 336: Annotation obtained for the genome of phage PMBT5 looks limited. Where HMM-based search used? Such as InterProScan. Likely additionally findings could be made if used.

Link for InterProScan: https://www.ebi.ac.uk/interpro/search/sequence/

Minor comments

  • Line 308: GC content of phage gDNA lower or higher than that of the host?

Round 2

Reviewer 2 Report

I sincerely congratulate the authors for the efforts they made using HHpred for ORFs' annoration. I think that was also benificial for them. 

The manuscript, as it is, is totally satisfactory.

Reviewer 3 Report

I am satisfied with the actions taken to address the issues I highlighted in my previous manuscript review.